# European Markets for Cultured Meat: A Comparison of Germany and France

**DOI:** 10.3390/foods9091152

**Published:** 2020-08-21

**Authors:** Christopher Bryant, Lea van Nek, Nathalie C. M. Rolland

**Affiliations:** 1Department of Psychology, University of Bath, Bath BA2 7AY, UK; 2Ipsos, 10115 Berlin, Germany; lea.vannek@ipsos.com; 3AgroSup, Université Bourgogne Franche-Comté, 25000 Dijon, France; ncm.rolland@gmail.com

**Keywords:** meat alternatives, cultured meat, consumer behaviour, attitudes, Europe

## Abstract

The negative impacts of meat consumption for animals, the environment, and human health are more pressing than ever. Although some evidence points to an ongoing reduction in meat consumption in Europe, consumers are overall unwilling to cut their meat consumption in a substantial way. The present study investigates dietary identities and perceptions of cultured meat in nationally representative samples from Germany (*n* = 1000) and France (*n* = 1000). Participants were recruited through an Ipsos panel to answer an online survey, which included questions about their current and intended consumption of conventional meat, as well as questions about their opinions of cultured meat. We find that, whilst rates of vegetarianism were relatively low in France, unrestricted meat-eaters were a minority in Germany, and concern for animal welfare was the most common reason given for meat reduction. Substantial markets for cultured meat exist in both countries, although German consumers are significantly more open to the concept than the French. Strikingly, cultured meat acceptance is significantly higher amongst agricultural and meat workers, indicating that those who are closest to existing meat production methods are most likely to prefer alternatives. We found some evidence that pro-cultured meat messages, which focus on antibiotic resistance and food safety, are significantly more persuasive than those that focus on animals or the environment. Furthermore, consumers project that they would be significantly more likely to consume cultured meat that does not contain genetically modified ingredients. Overall, we find substantially large markets for cultured meat in Germany and France, and identify some potential ways to further increase acceptance in these markets. We conclude by highlighting the most promising markets for cultured meat, and highlighting a lack of antibiotics as a potentially persuasive message about cultured meat.

## 1. Introduction

### 1.1. Meat in Europe

It is becoming increasingly clear that our current methods of intensive animal farming in Europe are unsustainable. As well as being a major contributor to greenhouse gas emissions [1,2] livestock production systems occupy 28% of land in the EU and contribute disproportionately to soil acidification, water pollution, and air pollution [3]. Around the world, industrial animal agriculture is a major driver of deforestation, antibiotic resistance, and zoonotic infections [4,5,6,7]. This confluence of negative externalities has resulted in recent calls to drastically curb our meat consumption [8,9]—however, most meat-eaters have no such intention [10]. Indeed, as populations in developing countries increase their purchasing power, global production of meat is forecast to increase by 15% in the decade to 2027 [11].

Per capita meat consumption currently is far higher in wealthier nations [12] and many of us are consuming far more meat than is optimally healthy [9]. It is therefore incumbent on Western consumers and policymakers to find ways to reduce meat consumption. Meat consumption has been trending slightly down in Germany [13] and in France [14], but remains far higher than the developing world, and therefore represents more potential for displacing demand.

However, reducing meat consumption in Europe is not straightforward. Not only are consumers largely unwilling to reduce their meat consumption [10], many may actually see eating meat as a central part of their social identities and traditions [15]. Family events, celebrations and rites of passage frequently involve communal consumption of meat, and for these reasons, targeting meat consumption is likely more emotive and more difficult than targeting other health and environmental behaviours.

Moreover, meat consumption in Europe is inextricably linked to thoroughly intertwined economic and political issues. In particular, Germany has historically had an agricultural deficit whilst neighbouring France is Europe’s largest agricultural producer, accounting for around 30% of agricultural output in Europe [16]. This has led to a situation where both of these powerful European nations are staunchly in support of the Common Agricultural Policy, a programme that is responsible for almost 40% of the EU budget [17] and effectively guarantees large subsidies to farming, including animal farming. Owing to strong support from agricultural interests in France, this is a policy that has proved particularly stubborn against reform [18]. Therefore, France and Germany are both vitally important countries to study in the context of European agriculture.

Political institutions and consumer preferences mean that reducing meat consumption in Europe is indeed a substantial challenge. Increasingly, research has looked to ways we can, and may be able to in the future, instead displace meat consumption.

### 1.2. Cultured Meat

One proposed solution to this predicament is cultured meat—real animal meat grown from cell cultures rather than as part of a living animal [19]. Advocates of the technology posit that, as well as circumventing the need to slaughter animals, cultured meat will have a much lower environmental footprint than meat from animals, largely due to its relative input efficiency [20]. Additionally, automated cultured meat production systems will enable us to minimise contact between humans and potentially-sick animals. This will drastically reduce the risk of zoonotic pathogens in the food system.

However, consumer acceptance of cultured meat is not universal [21]. Although some surveys have shown that the majority of US consumers would eat cultured meat [22,23], cross-country data indicates that acceptance is likely lower in Europe compared to the US [24], a trend anticipated by experts in the field [25].

Several published studies have investigated cultured meat acceptance in Europe; however, the acceptance rates they report are not necessarily comparable due to different question design and different sampling methods. Mancini and Antonioli [26] reported that 54% of their Italian sample said they would try cultured meat, whilst Flycatcher [27] found that 52% of Dutch consumers said they would eat it. Rolland, Markus and Post [28] found, in a simulated cultured meat tasting, that all participants offered a piece of meat labelled as ‘cultured meat’ ate it, and they tended to rate it as better-tasting than conventional meat despite a lack of objective difference. Eurobarometer [29] was the only source of cross-country data within Europe until recently: new research tentatively finds higher acceptance in Spain compared to the UK [30], while data on older consumers shows the UK and Spain about equal in terms of cultured meat acceptance, higher than Poland, but lower than Finland and the Netherlands [31].

France and Germany have both received relatively little attention in the research on cultured meat acceptance, yet both are centrally important to European food policy [32]. One study purported to show very low acceptance in France [33]; however, this study had several methodological limitations, including response options that were not mutually exclusive, and a significant over-representation of participants connected to the meat industry. A more recent study showed German acceptance at 57%, which appears more in line with other European surveys [34]. Further, Dupont and Fiebelkorn [35] found that German school children and adolescents were overall positive towards cultured meat, and most indicated they would prefer it over insects, despite perceiving insects as healthier and more natural.

This study sought to build on existing evidence to investigate attitudes towards cultured meat and related foods in France and Germany. In particular, we wanted to address the following research questions:What proportion of the German and French populations eat conventional meat?What are the overall rates of cultured meat acceptance in Germany and France?What differences exist in cultured meat acceptance between Germany and France?What demographic factors predict acceptance of cultured meat within Germany and France? What additional information is most effective at increasing cultured meat acceptance?Do consumers differ in their willingness to consume GM (genetically modified) vs. non-GM cultured meat?

## 2. Methods

### 2.1. Process

To answer these questions, we conducted an online survey of potential cultured meat consumers in France and Germany. The survey, administered by Ipsos in December 2019, used Computer-Assisted Web Interviews (CAWI) to ask questions about participants’ current diets as well as their attitudes towards cultured meat and other meat alternatives. As well as consumer analysis within each country based on preferences and demographic data, this design allows us to directly compare responses between each country.

The data for this study was collected before the involvement of any authors connected with academic institutions. As such, formal ethical approval of this study by an IRB was not sought or granted. However, as with all data collected by Ipsos, the process adhered entirely to the ESOMAR guidelines on ethical online research [36]. This includes assurances that participants gave informed consent to take part, were protected from any potential harms, and had their personal data protected.

The full survey instrument is provided in the Appendix A, but a narrative summary of the questions is given here. First, participants were asked some basic demographic questions about their gender, age, and region. Participants then answered some questions about their concern for various issues related to animal agriculture, including animal welfare, the environment, world hunger, and food safety. They then indicated which diet they identify with, their frequency of meat consumption, and whether they intended to reduce their meat consumption. Next, participants who indicated that they followed a meat-restricted diet, or intended to reduce their meat consumption, selected their reasons for this from a multiple selection list.

The next part of the survey asked about cultured meat. Participants answered questions about their familiarity with cultured meat, as well as their favorability of the concept. They then answered several questions about their intentions with respect to cultured meat, including their willingness to try it, buy it, and use it to replace conventional meat in their diet. This included questions about whether they would replace conventional meat with genetically modified cultured meat, and with non-genetically modified cultured meat.

Next, participants answered questions about various ‘pressure points’ relating to different guarantees about cultured meat. They indicated whether their motivation to consume cultured meat would increase with additional information that cultured meat is better for the environment, avoids killing animals, reduces the risk of food-related pathogens, and avoids using antibiotics. Finally, participants indicated their preference between cultured meat and plant-based meat, and between cultured meat and insect protein.

Participants then answered some final demographic questions, indicating their level of education, employment status, political views, income, and town size. As well as these standard demographic questions, participants indicated whether they currently work in the agricultural or meat production industries. This was thought to be potentially relevant to participants’ opinions of meat, cultured meat, and related issues, especially since concern for agricultural workers is a common theme in some consumer acceptance literature [21].

### 2.2. Participants

Participants were recruited through the Ipsos Digital panel. The sample of 2000 adults (1000 each in France and Germany) was representative of working age adults (aged 18–65) according to age and gender quotas (see Table 1).

Participants gave their informed consent before taking part in the study, and were compensated for their participation through Ipsos’ system of incentive points, which can be exchanged for vouchers or donations. The sample demographics are shown in Table 1.

### 2.3. Statistical Analyses

In order to answer the research questions, we used a variety of statistical techniques.

Our first question about conventional meat consumption in France and Germany can be answered using descriptive data primarily. As well as the percentages of participants identifying with each diet, we reported the percentages intending to reduce their meat consumption in the future, and the reasons given by those reducing or intending to reduce their meat consumption. In order to test statistically for differences in the rates of strict vegetarianism and veganism between Germany and France, we also recoded the dietary variable to binary ‘vegetarian’ and ‘vegan’ variables for use in 2 × 2 chi square analyses. Differences in the proportion of vegetarians and vegans between the two countries would be considered significant if *p* < 0.05.

Our second research question about overall cultured meat acceptance in France and Germany is similarly answered using descriptive data primarily. We reported the percentage of respondents in each country answering on the positive side, the negative side, or the centre of the 5-point Likert scales for a variety of questions about cultured meat. This includes purchase intent, willingness to try, and familiarity. We also reported the percentages of respondents with a preference for cultured meat or other alternative proteins (plant-based meat and insects) in each country.

Our third research question about differences in cultured meat acceptance between Germany and France was addressed using a series of independent *t*-tests. We compared responses to five different questions about cultured meat acceptance. In order to account for the effect of multiple comparisons increasing the chances of false positives, differences were considered significant at a Bonferroni-corrected *p* < 0.01 [37].

Our fourth research question about demographic factors associated with cultured meat acceptance was addressed using multivariate linear regressions. We ran one regression for each country; cultured meat purchase intent was used as the dependent variable, and demographic factors were entered as predictor variables. Gender, age, town size, political views, income group, diet, and whether the respondent was an agricultural worker, were all entered as predictor variables. The overall regression models were considered significant if *p* < 0.05 for the F value, and predictor variables were considered significant if *p* < 0.05 for the β value.

Our fifth research question about what additional information is most effective at increasing cultured meat acceptance was addressed using a series of paired *t*-tests to compare responses to each of the four guarantees. All meat-eating participants rated all four pieces of information (i.e., this was not manipulated experimentally). Differences between the mean values for each pair were considered significant at *p* < 0.05.

Finally, our sixth research question about consumer acceptance of GM vs. non-GM cultured meat was addressed using paired *t*-tests within countries. Again, all meat-eating participants rated their likelihood of consuming both GM and non-GM cultured meat, making an independent *t*-test inappropriate. Differences between the ratings for GM and non-GM cultured meat were considered significant where *p* < 0.05.

## 3. Results

In this section, we report six sets of analyses, which correspond to our six research questions.

### 3.1. Dietary Identities

We observed the number of people who identified with each of the below dietary labels (Figure 1).

Strikingly, the majority of German respondents did not identify as omnivorous, and large portions of both countries identified as flexitarian, indicating growing interest in deliberate meat reduction. Conversely, in France, over two thirds indicated that they were omnivorous, with relatively few strict vegetarians and vegans, though around one quarter of the French sample identified as flexitarian.

Further, we compared rates of vegetarianism and veganism between each country. In order to do this, we combined these dietary categories to code all participants as vegetarian or non-vegetarian and vegan or non-vegan. We then compared these binary variables across the countries in a 2 × 2 chi square analysis. Note that in this analysis, vegans are included in both the vegetarian and the vegan rate, since vegans are a subset of vegetarians.

Our analyses showed that vegetarianism in Germany (6.5%) was significantly higher than vegetarianism in France (2.6%) (χ^2^ (1) = 17.511, *p* < 0.001), and that veganism in Germany (1.9%) was significantly higher than veganism in France (0.6%) (χ^2^ (1) = 6.846, *p* = 0.009).

In addition, participants who indicated that they currently eat meat were asked about their intentions to reduce their meat consumption in the future. This data is shown in Figure 2.

As shown in Figure 2, of those who ate meat, almost 50% intended to reduce their consumption in Germany and France. Just 35.5% of non-reducers in Germany said they did not intend to reduce, and this number was just 31% in France. This shows a positive forecast towards future meat reduction.

Those following diets that limit meat consumption, or intending to reduce their meat consumption, were also asked about their main reasons for doing this. Respondents could select multiple reasons. These are presented in Figure 3. Unfortunately, due to a programming error in the survey instrument, the ‘environment’ offer was not shown to participants in the French survey, and therefore this data is missing. Since participants had to select at least one answer, the missing option here likely led to increases in the selection of other options in the French sample. For example, 36.5% of the French respondents indicated that price was an important concern, compared to just 16.0% of German respondents, and 10.3% selected ‘other’ compared to just 5.3% in the German sample. Although French meat-reducers who were concerned with the environment could have chosen the ‘Other’ option when they saw that environment was not offered as an option in the list, we note that survey respondents likely tend not to do this when their answer is not offered as one of the main list. Therefore, it is likely that the omission of the ‘Environment’ answer in France led to increases in other responses, notably that meat is expensive.

As shown, a concern for animals was the most common reason given by German participants, slightly more common than a concern for the environment. Due to missing data, we cannot make this comparison in France, although the order of the other variables is very similar, so we can infer that the environment is likely to be one of the top concerns alongside animal welfare for meat-reducers in France as well as Germany.

### 3.2. Overall Cultured Meat Acceptance

Overall, we found evidence for substantial interest in cultured meat in France and Germany despite most participants not having heard of it before (see Table 2). In France, respondents tended to be against cultured meat more than in favour, but were split quite evenly on the question of whether they would try it or buy it themselves. In Germany, respondents were very enthusiastic about cultured meat—the majority said they would buy cultured meat when it is available, and the majority of meat-eaters said they would use cultured meat to replace conventional meat. Further, measures of acceptance were significantly higher in Germany compared to France (see Table 3).

Further, participants indicated their preferences between cultured meat and insects, and between cultured meat and plant-based meat. The results shown in Figure 4 and Figure 5 only show participants who indicated a preference (i.e., not those who said they would eat neither, both, or don’t know).

As shown, German respondents tended to prefer cultured meat over plant-based meat, whilst French respondents were quite split between the two. Comparisons with eating insects, on the other hand, tended to favour cultured meat strongly in both countries.

### 3.3. Country Differences in Cultured Meat Acceptance

We used a series of independent samples *t* tests to compare perceptions of cultured meat between Germany and France. As discussed in the methods section, significance here is achieved at *p* < 0.01 to account for multiple comparisons [37].

The results showed significantly higher familiarity with, more positive perceptions of, and higher intentions to consume cultured meat in Germany compared to France, as shown in Table 4.

### 3.4. Predictors of Cultured Meat Acceptance

We ran multivariate linear regressions within each country using demographic variables as predictors and cultured meat purchase intent as the outcome variable. Frequency of meat consumption, age group, rurality, political views, income group, and diet were entered as scale predictors, whilst gender and being an agricultural worker were entered as dummy (binomial) variables. Although the regression models had low *R*^2^ values, they still reached significance and identified multiple significant predictors of cultured meat acceptance in each country. This type of result indicates that although some demographics tend to be more open to cultured meat, there is still a lot of variation within each group.

Factors predicting higher acceptance in France are being male, being younger, living in a more urban area, working in animal agriculture or meat production, and marginally by higher meat consumption (*p* = 0.079). In Germany, higher acceptance was predicted by being younger, and marginally by working in animal agriculture or meat production (*p* = 0.070). Whilst age, gender, and urbanness have predicted cultured meat acceptance in previous studies [21,38,39,40], the finding that those who work in animal agriculture or meat production tend to be more accepting of cultured meat is a novel and surprising one.

### 3.5. Increasing Cultured Meat Acceptance

In order to investigate the impact of different messages on cultured meat acceptance, we asked four questions about respondents’ projected attitudinal change in the light of new information. Participants could indicate that each piece of information probably/definitely would/not motivate them to consume cultured meat. Note that these motivations were self-reported by all participants (i.e., this was not tested experimentally).

As shown in Figure 6, responses were again more positive in Germany overall, but the pattern of results is similar: an assurance that cultured meat is free from antibiotics appeared to be more persuasive than assurances about animal welfare, the environment, or food pathogens.

To test whether these messages resulted in significantly different projected acceptance rates, we used a series of paired *t* tests to compare the aggregated responses to each of the four messages (see Table 5). We also provide the mean values for each motivator for ease of comparison. In the first row, lack of shared subscript letters indicates a significant difference (i.e., cells that share a letter do not differ significantly from each other).

As shown here, the pressure point that was most motivating was the potential for cultured meat to alleviate animal farming’s impact on antibiotic resistance. This was significantly more persuasive than food safety, which in turn was significantly more persuasive than the environment and animals.

### 3.6. GM vs. Non-GM Cultured Meat

We used paired *t*-tests to compare responses to two questions that asked about whether respondents would be willing to replace their meat consumption with GM or non-GM cultured meat. The results are shown in Table 6.

As shown, consumers in both countries were significantly more likely to replace conventional meat in their diet with cultured meat if it is non-GM.

## 4. Discussion

The present study reports a multitude of novel and important findings with respect to cultured meat acceptance in Europe. Here, we offer some interpretations of the findings, as well as their implications for cultured meat researchers, advocates, and policymakers.

First, one of the most striking findings of the study is a minority of consumers identifying as unrestricted omnivores in Germany. Germany is known to be one of the most vegetarian nations in Europe [41], and per capita meat consumption has been trending down for several decades [12]. That said, surveys have shown that the large majority of Germans still eat meat [41]. For the first time, to our knowledge, we find that if you are living in Germany and you are not deliberately limiting your meat consumption, you are in a minority. The social implications of this could be profound: given the importance of normality as a central justification for meat consumption [42], the idea that eating as much meat as you want puts you in a minority is a powerful one for meat reduction advocates in Germany. We also find that almost half of meat-eaters in both Germany and France intend to reduce their consumption of meat, indicating further shifts in the future towards meat-free and meat-reduced diets.

With respect to cultured meat, we found evidence for substantially large markets in both France and Germany. Projected acceptance is significantly higher in Germany compared to France, and is significantly higher amongst younger consumers in Germany and France, as well as male and more urban consumers in France. These demographic factors have been associated with higher cultured meat acceptance in previous studies [21,22,38]. Moreover, the difference between the countries concurs with the difference we might expect based on each country’s agricultural makeup, as well as the difference observed in previous survey research [29].

One novel finding to come from the demographic data is the finding that people who work in animal agriculture or meat processing are significantly more likely to purchase cultured meat compared to those who do not. This is counterintuitive in the sense that one might expect agricultural workers to oppose a technology which may threaten to displace their job—indeed, this appears to be the basis for higher rejection of cultured meat by more rural consumers [38]. However, farmers may also see cultured meat as a way to address the mass demand for affordable meat, enabling them to move away from intensive industrial production systems and return to more traditional systems, which are more harmonious with environmental and animal welfare outcomes.

We also find some evidence that pro-cultured meat messages about antibiotic resistance and food safety are significantly more persuasive to European consumers than messages about animals and the environment. This may be because these issues are less well-known compared to issues of animal welfare and climate change [43], and therefore the information was more likely to be new. Indeed, previous research has found that consumers most commonly understand the benefits of cultured meat in terms of animals and the environment, whilst other benefits are less commonly perceived [21]. Moreover, these benefits may be interpreted as primarily accruing to the individual rather than society, a strategy that might counter consumers’ tendency to think of cultured meat in terms of individual risks and societal benefits [44]. Therefore, advocates should be mindful of promoting a broad range of messages, which ensures that benefits relevant to a range of consumers are salient.

Finally, we find evidence that genetically modified cultured meat (or cultured meat that contains genetically modified ingredients) is likely to be met with far more resistance from consumers compared to non-genetically modified cultured meat. Genetically modified crops are essentially banned in Europe currently [45], so it is perhaps unsurprising that the inclusion of these ingredients increases rejection in a European sample. Whilst many parallels exist in terms of the public perceptions of each technology [21,46], the increased rejection of genetically modified cultured meat may reflect a compounding effect whereby the presence of two unnatural features is perceived as worse than the sum of each unnatural feature present by itself. In other words, meat that is cell cultured and genetically modified is likely to be perceived as far worse than meat that is only cell cultured or only genetically modified.

Overall, we conclude that France and Germany represent viable markets for cultured meat. Germany has a high rate of vegetarianism and concern for related issues, while less than half of the population are unrestricted meat-eaters, representing a new normal for Germany with respect to meat consumption. Although these concerns are not as great in France, close to half of French meat-eaters intend to eat less meat in the future, meaning concern for these issues may be increasing. Over a third of the French, and a majority of Germans, would buy cultured meat. Acceptance is higher overall in Germany, though certain demographic groups typically have higher acceptance, highlighting potential markets for cultured meat. The assurance that cultured meat is free from antibiotics may represent a promising basis for promotion.

There are several limitations to consider in the present study. First, questions about different pro-cultured meat messages and GM vs. non-GM cultured meat were asked to all participants, rather than giving different options to different participants experimentally. Therefore, these responses only reflect participants’ projected preferences. For example, it is possible that consumers in fact would not respond very differently to genetically modified cultured meat in an experimental paradigm, but report that they would because the question makes genetic modification salient. A more reliable conclusion on this question will come from experimental work where participants are exposed to GM or non-GM cultured meat. Second, the omission of the ‘environment’ option in the French version of the survey meant that those who indicated they were currently reducing their meat consumption missed out on one of the major reasons given by the German sample. As we have discussed, this omission also likely led to an increase in the proportion of French respondents selecting other options instead. This omission only affected one question, which was only asked of respondents who were reducing or intending to reduce their meat consumption. Nonetheless, the data from France on this question (Figure 3) should be interpreted with considerable caution. Third, the study is subject to well-known methodological limitations of self-reported online surveys including social desirability bias and attention effects. Use of standardised Ipsos survey design and question wording sought to minimise these impacts.

Future research should further investigate the potential for increasing cultured meat acceptance by highlighting benefits that are less intuitive, or that explicitly benefit the consumer rather than society in general. Furthermore, experimental work should investigate the extent to which consumer perceptions of cultured meat do, in fact, vary with the addition of genetically modified components. Further cross country data can locate France and Germany alongside other European cultured meat markets to help producers identify the markets with the greatest potential.

## Figures and Tables

**Figure 1 foods-09-01152-f001:**
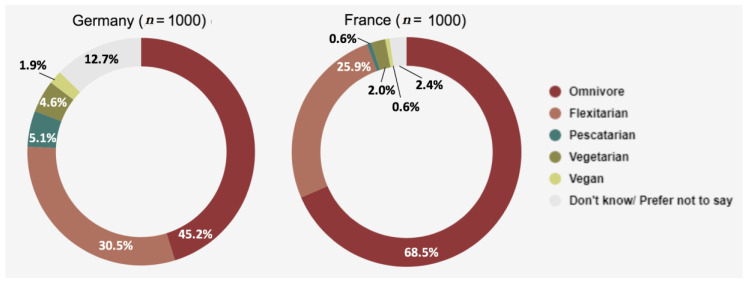
Dietary identities in Germany and France.

**Figure 2 foods-09-01152-f002:**
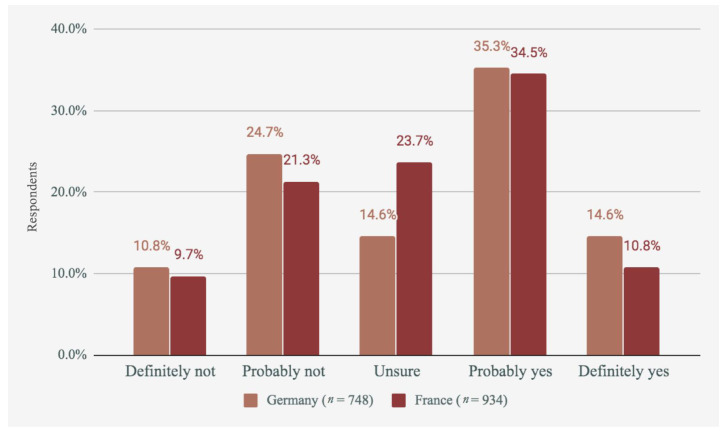
Intentions to reduce meat consumption in each country.

**Figure 3 foods-09-01152-f003:**
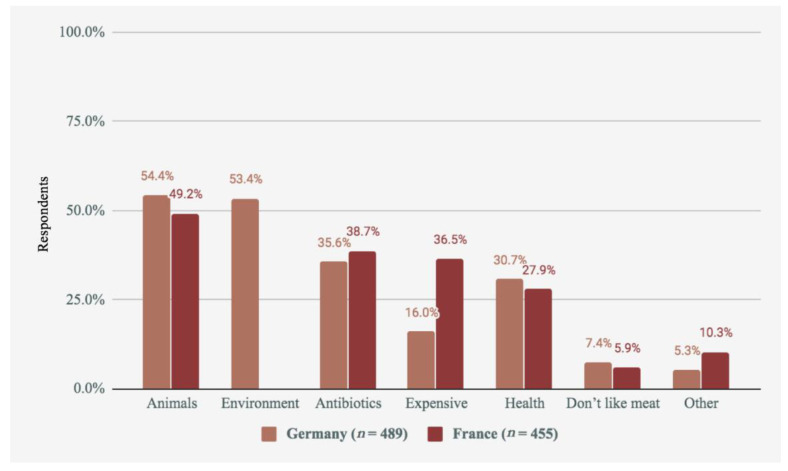
Reasons for reducing meat consumption reported in each country.

**Figure 4 foods-09-01152-f004:**
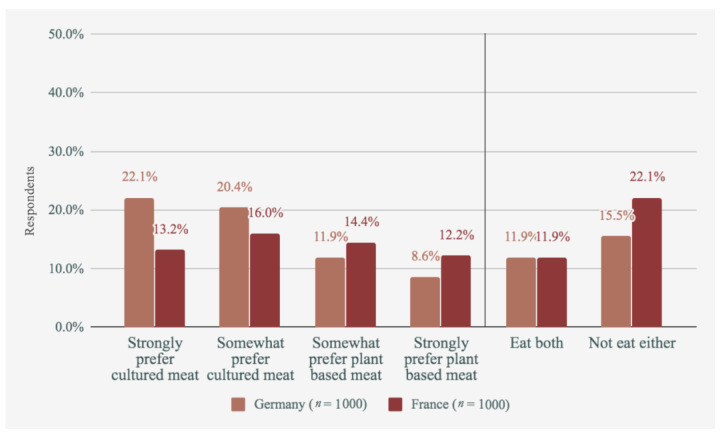
Cultured meat vs. plant-based meat.

**Figure 5 foods-09-01152-f005:**
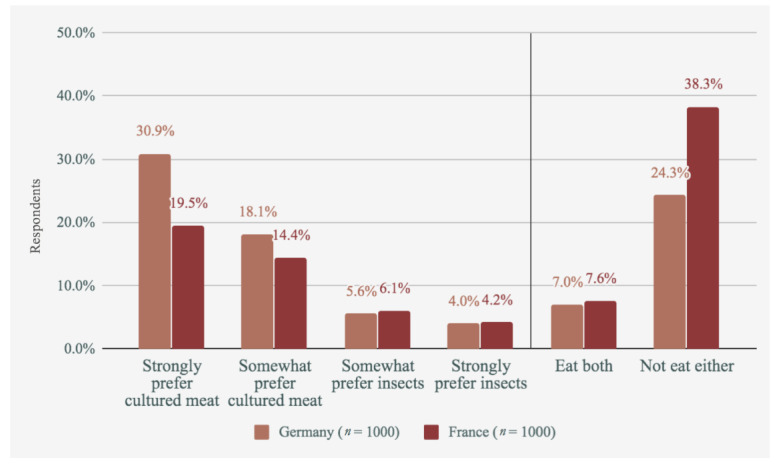
Cultured meat vs. insects.

**Figure 6 foods-09-01152-f006:**
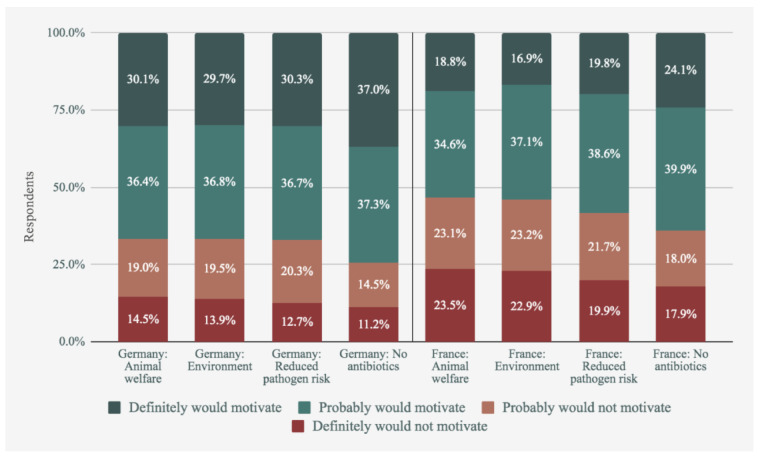
Attitudinal change towards cultured meat according to each piece of information.

**Table 1 foods-09-01152-t001:** Basic sample demographics.

		France (*n* = 1000)	Germany (*n* = 1000)
**Gender**	Male	47.4%	49.2%
	Female	52.6%	50.8%
**Age**	18–34	34.0%	33.0%
	35–54	45.0%	45.0%
	55–65	21.0%	22.0%

**Table 2 foods-09-01152-t002:** Familiarity with, favorability for, and consumption intention of cultured meat in Germany and France.

	Germany			France		
	Yes	Maybe	No	Yes	Maybe	No
**Heard of cultured meat ***	22.6%	21.9%	55.5%	16.0%	21.6%	62.4%
**In favour of cultured meat**	32.6%	39.7%	27.7%	20.1%	30.8%	49.1%
**Would try cultured meat**	58.3%	8.1%	33.6%	44.2%	9.8%	46.1%
**Would buy cultured meat**	55.7%	12.4%	31.9%	36.8%	13.8%	49.4%
**Would replace conventional meat (meat-eaters only)**	53.1%	12.8%	34.1%	34.2%	12.9%	52.8%

* For this question, the figure in the ‘maybe’ column corresponds to those who had heard of cultured meat but did not know what it was (see question wording in Appendix A).

**Table 3 foods-09-01152-t003:** Measures of cultured meat acceptance in Germany and France.

Measure	Germany Mean (SD)	France Mean (SD)	*T*-test
Familiarity	1.67 (0.82)	1.54 (0.76)	t = 3.676, *p* < 0.001 *
In favour	3.02 (1.21)	2.41 (1.24)	t = 10.788, *p* < 0.001 *
Willing to try	2.71 (0.97)	2.33 (1.02)	t = 7.984, *p* < 0.001 *
Willing to buy	2.67 (0.97)	2.25 (1.01)	t = 8.905, *p* < 0.001 *
Replace	2.61 (1.01)	2.15 (1.00)	t = 8.796, *p* < 0.001 *

* indicates that the difference between the countries is statistically significant.

**Table 4 foods-09-01152-t004:** Regression models predicting cultured meat purchase intent in Germany and France.

	Germany		France	
	*R*^2^ = 0.045, Adj *R*^2^ = 0.031 F (8550) = 3.245, *p* = 0.001		*R*^2^ = 0.068, Adj *R*^2^ = 0.056 F (8592) = 5.424, *p* < 0.001	
	Standardised *β*	*p*	Standardised *β*	*p*
(Constant)	3.564		2.924	
Frequency of meat consumption	−0.006	0.893	0.078	0.079
Gender (female)	−0.024	0.590	−0.094 *	0.021
Age group	−0.135 *	0.002	−0.167 *	<0.001
Urbanness	−0.016	0.710	0.086 *	0.033
Agricultural worker	0.081	0.070	0.100 *	0.014
Political views	−0.056	0.196	−0.025	0.546
Income group	−0.048	0.266	−0.018	0.653
Diet	0.072	0.115	−0.021	0.643

* indicates that this variable significantly predicted cultured meat purchase intent.

**Table 5 foods-09-01152-t005:** Comparison of the impact of different motivators on cultured meat purchase intent.

	Environment	Animals	Food Safety	Antibiotics
**Mean (SD)**	2.65 (1.04)_a_	2.64 (1.05)_a_	2.71 (1.02)_b_	2.84 (1.02)_c_
Comparisons (*t* tests)				
**Environment**	-	-	-	-
**Animals**	t = 0.00, *p* = 1.00	-	-	-
**Food Safety**	t = 3.94, *p* < 0.01 *	t = 4.35, *p* < 0.01 *	-	-
**Antibiotics**	t = 11.55, *p* < 0.01 *	t = 11.36, *p* < 0.01 *	t = 9.04, *p* < 0.01 *	-

Differing subscript letters indicate that these results were significantly different (i.e., if they do not share a subscript letter). * indicates that the pairwise comparison between the row variable and the column variable using a *t* test was significant.

**Table 6 foods-09-01152-t006:** Paired *t*-tests indicating a strong preference for non-GM over GM cultured meat.

	Non-GM Mean (SD)	GM Mean (SD)	*t*	*p*
Germany *	3.22 (0.58)	2.46 (0.91)	−14.255	<0.001
France *	3.19 (0.60)	1.94 (0.94)	−19.292	<0.001

* indicates that there was a significant difference between projected acceptance of GM and non-GM cultured meat for this country.

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
