# Peer review of "European Markets for Cultured Meat: A Comparison of Germany and France"

_foods, 2020, doi:10.3390/foods9091152_

Round 1

Reviewer 1 Report

Foods-889339

The work conducted in this manuscript is very timely as the agenda of alternative proteins are moving in the direction of including the concept of cultured meat. Consumers attitude and readiness for this product is therefore very relevant to investigate and some new and interesting results has been revealed.

General comments:

The technical error leading to the missing option of selecting “environmental” in the French study is very unfortunate and ought to be addressed a bit more when describing the results –see comments related to Graph 2 results below.  

Legends to graphs and tables should be more informative e.g. including number of respondents (n=1000), in graphs y-axes should indicate “respondents”. Abbreviations should be spelled out e.g. CM and PBM in figure 3 and 4

Specific comments and suggested changes:

Line 16: change “animal welfare the most” to “animal welfare as the most”

Line 37: delete space before “Jones”

Line 45: could “for our own good” be changed to “for health reasons”

Line 65: change “this a policy” to “this is a policy”

Lines 175-176: vegetarianism in Germany given as 7.4% but when adding the numbers of vegetarian and vegan in Graph 1 it is only 6.6 (1.9 + 4.6). One or the other should be corrected to align Graph 1 and the text. The same goes for veganism in Germany where the graph show 1.9% and the test say 2.2%.

Line 191/Graph 2 results: This section should include a comment on the large difference between countries regarding the category “expensive”. The category of “other” is double in France compared to Germany. This should be commented and possibly linked to the missing option of “environment” in the questionnaire in France.

Line 198: change “would try it themselves“ to “would try it or buy it themselves”

Line 200: is missing a reference to the percentage in France... This could be “ In France only one third would replace conventional meat with cultured meat”.

Table 4 (and text describing this table): What does PP mean? This should be in the legend to the table. It is not clear where these data originate from. The questionnaire indluced in the manuscript does not seem to support these data… How can PP Environment be evaluated for France when that term was accidently omitted in the French version? How can food safety and antibiotics be evaluated separately when they are combined in the questionnaire (if data is from the described queationnaire). Description of data related to the last half of the table (PP Environment, PP animals, PP Food safety and PP Antibiotics) needs to be described and interpreted in context.

Graph 5: why is France and Germany combined? If I is because here was no difference this should be mentioned - or any other reason should be given. It should be mentioned that “environment”- data is from Germany only

Line 303: you could add “to our knowledge” after “For the first time”

Line 305: the impact of the tipping point regarding “normality and who is the minority” is very interesting but maybe the word “huge” is too strong taking into account that that the tipping point is just reached? Wording could be “considerable” or something in this direction….?

Line 360. At the end of this paragraph is would be appropriate to mention some of the obvious limitations like samples size for the study and possible consequences of the intentional missing “environmental” term in the French version.

Author Response

We are grateful to the reviewer for taking the time to review the manuscript. We have made many changes based on the reviewer’s comments, and we are confident that this has much improved the manuscript. We detail these changes in the table below with respect to the reviewer’s comments.

Comment

Response

The technical error leading to the missing option of selecting “environmental” in the French study is very unfortunate and ought to be addressed a bit more when describing the results –see comments related to Graph 2 [now Graph 3] results below.

Indeed, we recognise that this is a limitation affecting the data for that question. We have added further discussion of this in line with your comments about Graph 2 [which is now Graph 3].

Legends to graphs and tables should be more informative e.g. including number of respondents (n=1000), in graphs y-axes should indicate “respondents”. Abbreviations should be spelled out e.g. CM and PBM in figure 3 and 4 [now 4 and 5]

We have added sample sizes and y-axis labels to all Graphs, and removed the abbreviations in Graphs 3 and 4 [now Graphs 4 and 5].

Specific comments and suggested changes:

Line 16: change “animal welfare the most” to “animal welfare as the most”

Line 37: delete space before “Jones”

Line 45: could “for our own good” be changed to “for health reasons”

Line 65: change “this a policy” to “this is a policy”

We are grateful to the reviewer for taking the time to identify specific typos in the manuscript. We have changed the wording in all the parts suggested.

Lines 175-176: vegetarianism in Germany given as 7.4% but when adding the numbers of vegetarian and vegan in Graph 1 it is only 6.6 (1.9 + 4.6). One or the other should be corrected to align Graph 1 and the text. The same goes for veganism in Germany where the graph show 1.9% and the test say 2.2%.

We thank the reviewer for noticing this error. This was caused by a failure to count ‘Don’t know/prefer not to say’ responses when converting from the survey responses to binary ‘vegetarian’ and ‘vegan’ variables for chi square analyses. We have now corrected this variable in the data by counting these responses as non-vegetarian and non-vegan, and updated this paragraph to reflect the correct numbers, which now concur with Graph 1.

Line 191/Graph 2 [now Graph 3] results: This section should include a comment on the large difference between countries regarding the category “expensive”. The category of “other” is double in France compared to Germany. This should be commented and possibly linked to the missing option of “environment” in the questionnaire in France.

We have added a paragraph explaining the possible effect of the missing option, as suggested. Thank you for your feed back on this point; it is prudent to advise readers to interpret results to this question with some caution in the French sample.

Line 198: change “would try it themselves“ to “would try it or buy it themselves”

We have made this change.

Line 200: is missing a reference to the percentage in France... This could be “ In France only one third would replace conventional meat with cultured meat”.

We have added a line to describe this result.

Table 4 (and text describing this table): What does PP mean? This should be in the legend to the table. It is not clear where these data originate from. The questionnaire indluced in the manuscript does not seem to support these data… How can PP Environment be evaluated for France when that term was accidently omitted in the French version? How can food safety and antibiotics be evaluated separately when they are combined in the questionnaire (if data is from the described queationnaire). Description of data related to the last half of the table (PP Environment, PP animals, PP Food safety and PP Antibiotics) needs to be described and interpreted in context.

We have provided the full survey instrument here to improve clarity on the questions used. That said, we have also removed the PP variables from this table, since these referred to the motivators which are already analysed separately (Graph 6). Hopefully this is clearer now.

Graph 5 [note this is now Graph 6]: why is France and Germany combined? If I is because here was no difference this should be mentioned - or any other reason should be given. It should be mentioned that “environment”- data is from Germany only

We have now separated this into each country. This data refers to a different question than the data in Graph 3 - that question was about reasons for reducing meat consumption, whereas this question was about whether these guarantees about cultured meat would motivate them to eat it. For this question, there was no missing data.

Line 303: you could add “to our knowledge” after “For the first time”

Added.

Line 305: the impact of the tipping point regarding “normality and who is the minority” is very interesting but maybe the word “huge” is too strong taking into account that that the tipping point is just reached? Wording could be “considerable” or something in this direction….?

We have changed the wording to ‘The social implications of this could be profound’ to reflect the future/speculative nature of the claim, and to make this sentence more formal.

Line 360. At the end of this paragraph is would be appropriate to mention some of the obvious limitations like samples size for the study and possible consequences of the intentional missing “environmental” term in the French version.

We have added a section to the limitations further discussing the omission of the ‘environment’ option in France. We believe the sample size (1,000 in each of Germany and France) and representativeness was more than adequate for this study based on margin of error estimates and sample sizes in studies of a similar nature. We are open to any specific concerns about the samples used.

Reviewer 2 Report

The paper raised an interesting question and the need for the study is quite well justified, although it could be improved somewhat.There are some strange things in the text, such as the fact that it has funding but was written before the authors were linked to any institution. It can also be an English problem, which should be reviewed along the whole paper. However, the biggest problem is the methodology.The survey design is not at all clear, and I sincerely believe that it has serious flaws. In addition, the sociodemographic variables used (such as being a farmer) have not been specified in the material and methods and are explained later in the results. As they have not been put into material and methods, the need to include any of them has not been explained in the introduction either. For example, why was being a farmer considered important, are there previous studies in that way? Even more serious is that the statistical analysis has not been explained. And, once at the results, it is clearly seen that the statistic was not correct, since discontinuous data is being treated as if it were continuous. Unless, of course, a conversion has been made in the data, which does not seem. With all that, I have not looked at the discussion and conclusions, it is not worth while the work has so many failures. I would recommend you: 1. explain and clearly show the entire survey questionnaire. 2. explain the done statistics and if you have made an inappropriate statistic, remake it. If necessary, seek the help of a mathematician. 3. Once the article is written, send it to an English reviewer. The matter is interesting, it is worth making the effort. Cheer up.

Author Response

We are grateful to the reviewer for taking the time to review part of the manuscript. We have made some changes based on the reviewer’s comments, and we detail these in the table below with respect to the reviewer’s comments.

Comment

Response

The paper raised an interesting question and the need for the study is quite well justified, although it could be improved somewhat.There are some strange things in the text, such as the fact that it has funding but was written before the authors were linked to any institution.

The study was funded by Aleph Farms, ProVeg, and the INNOV project at INRAE (see funding statement). We have added details of the authors’ associations with these organisations.

It can also be an English problem, which should be reviewed along the whole paper.

We have corrected some typos and unclear wording on the suggestion of other reviewers. We invite any specific examples of places where English language is an issue.

However, the biggest problem is the methodology. The survey design is not at all clear, and I sincerely believe that it has serious flaws. In addition, the sociodemographic variables used (such as being a farmer) have not been specified in the material and methods and are explained later in the results. As they have not been put into material and methods, the need to include any of them has not been explained in the introduction either. For example, why was being a farmer considered important, are there previous studies in that way?

We have improved the clarity of the methodology by removing Table 2, and instead including the whole survey instrument as an Appendix so that readers can see all original question wording. We have also added a narrative description of the questions included, including a justification for asking whether participants currently work in agriculture or meat production. We note that this variable was indeed predictive of attitudes towards cultured meat (see regression analyses). We invite the reviewer to identify any specific ‘serious flaws’, since none were identified here.

Even more serious is that the statistical analysis has not been explained. And, once at the results, it is clearly seen that the statistic was not correct, since discontinuous data is being treated as if it were continuous. Unless, of course, a conversion has been made in the data, which does not seem.

We have added a large section to the methods describing the statistical analyses used in detail (see Section 2.3).

We note that it is very common for survey research to analyse single Likert scales using parametric tests including t tests and multivariate regression. This is done in many studies in the cultured meat literature, including Dupont and Fiebelkorn (2020), Wilks et al. (2019), Gomez-Luciano et al. (2019), and Rolland et al. (2020) (all cited in the manuscript).

The reviewer’s comments on the t-tests have misunderstood the analysis. We are not looking for a difference between the observed value and an a priori determined value, but for differences between the values in Germany and France (hence there is one t test per row in Table 4).

With all that, I have not looked at the discussion and conclusions, it is not worth while the work has so many failures. I would recommend you: 1. explain and clearly show the entire survey questionnaire. 2. explain the done statistics and if you have made an inappropriate statistic, remake it. If necessary, seek the help of a mathematician. 3. Once the article is written, send it to an English reviewer. The matter is interesting, it is worth making the effort. Cheer up.

We are grateful to the reviewer for taking the time to review half of the paper.

1. We have included the entire survey questionnaire for clarity.

2. We have added a part about the statistical analyses in the methods section. We have addressed the issue of the specific statistical tests there, and we invite the reviewer to point to any specific points on which they have concerns.

3. The first author is a native English speaker. We note several errors in the reviewers’ English, and therefore politely decline to take their advice on this point. We will, of course, correct any specific errors that the reviewer can point to.

Reviewer 3 Report

The objective of this work is to deepen the acceptance of cultured meat in two very different countries such as France and Germany. This study shows very interesting results regarding the evolution of the markets in the medium term.

However, some things need to be improved on the paper.

First, the summary is only a summary of the discussion, it should contain indicative information of the rest of the sections of the paper, such as the number of surveys, methodology and some significant number of the results.
The number of keywords is excessive, I recommend reducing them.
Material and methods section. It is necessary to describe the statistical methodology that has been carried out with the surveys.
Results section. With respect to Table 3, it could be improved by indicating if the differences between the groups and the responses are statistically significant.
In some tables such as Table 6, indications are missing at the bottom of the table, for the correct interpretation of the information. As is the case of the letters that appear as subscripts.
It would be advisable to include some small conclusions at the end.
Finally, English must be revised.

Author Response

We are grateful to the reviewer for taking the time to review the manuscript. We have made many changes based on the reviewer’s comments, and we are confident that this has much improved the manuscript. We detail these changes in the table below with respect to the reviewer’s comments.

Comment

Response

First, the summary is only a summary of the discussion, it should contain indicative information of the rest of the sections of the paper, such as the number of surveys, methodology and some significant number of the results.

We have added a brief description of the methods in the abstract. The main section here describes the results (without reporting exact figures). We have also added a line about the conclusions.

The number of keywords is excessive, I recommend reducing them.

We have removed three of the more specific keywords so that there are now 5 in total.

Material and methods section. It is necessary to describe the statistical methodology that has been carried out with the surveys.

We have added an extensive description of the statistical methods used (section 2.3)

Results section. With respect to Table 3, it could be improved by indicating if the differences between the groups and the responses are statistically significant.

Table 3 provides descriptive statistics only, used to give an overview of the market size. Significant differences between the country responses for these variables are shown in Table 4.

In some tables such as Table 6, indications are missing at the bottom of the table, for the correct interpretation of the information. As is the case of the letters that appear as subscripts.

We have added table annotations for Tables 4, 5, 6, and 7 to provide further details of the data represented there.

It would be advisable to include some small conclusions at the end.

We have added a concluding paragraph which summarises the main points from the discussion (before addressing limitations and areas for future research)

Finally, English must be revised.

We have made some corrections to typos and wording based on feedback from another reviewer. We are happy to correct the language in any specific places the reviewer can identify.

Round 2

Reviewer 2 Report

The manuscript has been improved. Concerning the statistical, that many authors do parametric statistics for nonparametric data does not mean that it is well done. However, the material and methods have been greatly improved, so the explanation of the results can now be followed.

This manuscript is a resubmission of an earlier submission. The following is a list of the peer review reports and author responses from that submission.